



# Bayesian full-waveform inversion of tube waves to estimate fracture aperture and compliance

Jürg Hunziker[1], Andrew Greenwood[1,2], Shohei Minato[3], Nicolas D. Barbosa[4], Eva Caspari[1,2], and Klaus Holliger[1,5]

[1]Applied and Environmental Geophysics Group, Institute of Earth Sciences, University of Lausanne, Lausanne, Switzerland
[2]Chair of Applied Geophysics, Montanuniversität Leoben, Leoben, Austria
[3]Faculty of Civil Engineering and Geosciences, Department of Geoscience and Engineering, Technical University Delft, Delft, The Netherlands
[4]Department of Earth Sciences, University of Geneva, Geneva, Switzerland
[5]School of Earth Sciences, Zhejiang University, Hangzhou, China

**Correspondence:** Jürg Hunziker (jurg.hunziker@unil.ch)

**Abstract.** The hydraulic and mechanical characterization of fractures is crucial for a wide range of pertinent applications, such as, for example, geothermal energy production, hydrocarbon exploration, $CO_2$-sequestration, and nuclear waste disposal. Direct hydraulic and mechanical testing of individual fractures along boreholes does, however, tend to be slow and cumbersome. To alleviate this problem, we propose to estimate the effective hydraulic aperture and the mechanical compliance of isolated
5    fractures intersecting a borehole through a Bayesian Markov chain Monte Carlo (MCMC) inversion of full-waveform tube-wave data recorded in a vertical seismic profiling (VSP) setting. The solution of the corresponding forward problem is based on a recently developed semi-analytical solution. This inversion approach has been tested for and verified on a wide range of synthetic scenarios. Here, we present the results of its application to observed hydrophone VSP data acquired along a borehole in the underground Grimsel Test Site in the Central Swiss Alps. While the results are consistent with the corresponding evidence
10   from televiewer data and exemplarily illustrate the advantages of using a computationally expensive stochastic, instead of a deterministic, inversion approach, they also reveal the inherent limitation of the underlying semi-analytical forward solver.

## 1   Introduction

Tube waves are interface waves propagating along the borehole wall. They are sometimes also referred to as Stoneley waves,
15   but, as Daley et al. (2003) point out, Scholte waves might be more appropriate as tube waves propagate along a solid-liquid interface. Primary sources of tube waves are ground roll passing over the well head (e.g., Hardage, 1981) or body waves encountering open fractures intersecting the borehole (e.g., Minato and Ghose, 2017; Greenwood et al., 2019b). Secondary sources are the borehole tool itself (e.g., Hardage, 1981) as well as changes in borehole radius or in acoustic impedance within the borehole annulus (e.g., Greenwood et al., 2019b).



Various modeling approaches have been proposed to study the properties of tube waves. A number of analytical techniques to calculate the tube-wave velocity (e.g., Chang et al., 1988; Norris, 1990) as well as semi-analytical methods to simulate complete waveforms (e.g., Cheng and Toksöz, 1981) have been published. To properly reproduce the effects of the borehole environment in finite-difference simulations, one needs a grid refinement in the immediate vicinity of the borehole (e.g., Falk et al., 1996; Sidler et al., 2013). Alternatively, a combination of a semi-analytical solution to model the borehole and a finite-difference approach to model the heterogeneous embedding background medium can be employed (e.g., Kurkjian et al., 1994).

As tube waves propagate along the borehole, no geometrical spreading occurs and, therefore, tube waves are much less attenuated than body waves and retain high amplitudes even at large distances from the source. Thus, if vertical seismic profiling (VSP) data are recorded with pressure sensors, such as hydrophones, tube waves tend to pose a problem as they cover body-wave reflections (e.g., Greenwood et al., 2019a, b). Without suppression or removal of the tube waves, reflections in hydrophone VSP data can, in general, only be interpreted at large source-receiver distances and then only before the tube waves and their reverberations arrive (Coates, 1998). Suppression of tube waves during data acquisition is discussed, for example, by Hardage (1981), Daley et al. (2003) and Greenwood et al. (2019b), amongst others. Methods to remove tube waves during data processing are proposed, for example, by Hardage (1981), Herman et al. (2000), and Greenwood et al. (2019a).

Here, we do not aim at suppressing or removing tube waves, but rather consider them as signals containing valuable information for characterizing hydraulically open fractures along the borehole, which is important for a wide variety of applications, such as groundwater management, geothermal energy production, hydrocarbon exploration, $CO_2$-sequestration, and nuclear waste disposal. If a tube wave is generated at a fracture due to an incoming P-wave, the amplitude ratio of the two wave types can be used to estimate fracture compliances (e.g., Bakku et al., 2013) or fracture permeability (e.g., Hardin and Toksöz, 1985; Li et al., 1994), while the amplitude ratio of P-wave-caused tube waves to S-wave-caused tube waves can be inverted for the orientation of fractures (e.g., Lee and Toksöz, 1995). The algorithm of Hornby et al. (1989) uses the arrival times of reflected tube waves to invert for the locations of permeable fractures and the reflectivity of tube waves to estimate the effective aperture of fractures. In the field of seismoelectrics, Zhu et al. (2008) showed that tube waves create electromagnetic waves when encountering fractures, which also have the potential to be used for fracture characterization.

The above methods do, however, require extensive manual conditioning of the data, like amplitude picking or time-gating of events. Furthermore, they are unable to provide an estimate of uncertainty and/or to identify multiple solutions that are equally likely. The objective of this work is to alleviate these limitations by providing an algorithm that considers the entire wavefield for characterizing fractures in terms of their hydraulic apertures and mechanical compliances as well as the associated uncertainties with a minimal amount of human interaction. To this end, we propose a Bayesian full-waveform inversion approach in combination with a recent semi-analytical approach (Minato and Ghose, 2017; Minato et al., 2017) as an efficient and robust forward solver. The proposed algorithm uses as input the complete P- and tube-wave fields with as little as possible preprocessing to invert for the effective hydraulic fracture aperture, the mechanical fracture compliance, the bulk- and shear-modulus of the background rock as well as some auxiliary parameters. We use a stochastic inversion algorithm in order (1) to obtain an entire ensemble of solutions, which, in turn, provides a measure of uncertainty and (2) to account for the strong non-linearity of the problem and to avoid getting stuck in local minima. We first present our stochastic full-waveform inversion approach,





followed by an application to field data from the Grimsel test site (www.grimsel.ch) in Switzerland and a subsequent discussion of the results.

## 2 Method

The goal of our stochastic inversion approach is to estimate the posterior probability density function (PDF) $p(\mathbf{m}|\mathbf{d})$, which in stochastic terms describes the adequacy of a model $\mathbf{m}$ given the observed data $\mathbf{d}$. We do this by relying on the following

approximation of Bayes' theorem (Bayes, 1763):

$$p(\mathbf{m}|\mathbf{d}) \propto p(\mathbf{m})L(\mathbf{m}|\mathbf{d}), \tag{1}$$

where $p(\mathbf{m})$ is the prior PDF describing any a priori knowledge we have about the model parameters and $L(\mathbf{m}|\mathbf{d})$ is the likelihood quantifying how well a model $\mathbf{m}$ explains the data $\mathbf{d}$. Following Tarantola (2005), we define the likelihood as:

$$L(\mathbf{m}|\mathbf{d}) = \frac{1}{(2\pi)^{D/2}\sigma_e^D} \exp\left(-\frac{1}{2\sigma_e^2}\sum_{j=1}^{D}(G_j(\mathbf{m}) - d_j)^2\right), \tag{2}$$

where $D$ and $\sigma_e$ are the amount of data points and the standard deviation of the data-error, respectively. The forward operator $G$ calculates synthetic data $\mathbf{d}^{\text{syn}}$ based on a model $\mathbf{m}$:

$$\mathbf{d}^{\text{syn}} = G(\mathbf{m}). \tag{3}$$

We use a novel semi-analytical algorithm for $G$, which evaluates the Green's function analytically in the frequency-space domain for a zero-offset VSP setting (Minato and Ghose, 2017). This is done in parallel for a limited number of individual

frequencies. Then, Green's functions for the complete frequency band are obtained by spline interpolation. The frequencies, for which the Green's functions are actually calculated, are selected such that the maximum error caused by the interpolation (i.e., the difference between an interpolated and a fully calculated dataset) is two orders-of-magnitude smaller than the largest value in the dataset. After a multiplication with the Fourier transform of the source wavelet and a subsequent inverse Fourier transformation, we obtain the full-waveform signals in the time-space domain.

In the considered forward operator $G$, seismic tube waves are generated and scattered at fractures characterized by their static apertures $L_0$ and compliances $Z$. A tube wave is generated when a P-wave hits a fluid-filled fracture intersecting the borehole, as the fracture is compressed and fluid is injected into the borehole. We describe this process in the frequency domain for a horizontal fracture with the tube-wave generation potential $\phi_g$:

$$\phi_g(z) = \sum_{i=1}^{N} \frac{2}{\rho_f c_T} \frac{p_t^{(i)}}{p_{\text{inc}}^{(i)}} \delta(z - z_i), \tag{4}$$

where $N$ is the number of fractures in the medium, $\rho_f$ the density of the fluid and $\delta$ the Dirac delta function. Depth is denoted by $z$ and sub- or superscripts $i$ refer to the $i$th fracture. The tube-wave velocity $c_T$ is given by (White, 1983):

$$c_T = \sqrt{\rho_f\left(\frac{1}{K_f} + \frac{1}{\mu}\right)^{-1}}, \tag{5}$$



with $K_f$ and $\mu$ being the fluid bulk modulus and the shear modulus of the formation, respectively. The pressure fields of the tube wave $p_t^{(i)}$ and the incoming P-wave $p_{\text{inc}}^{(i)}$ are then given by:

$$p_t^{(i)} = \sigma_0 \frac{j\omega c_T}{k_r \alpha_f} \frac{\rho_f Z \alpha_{\text{eff}}}{R} \frac{H_1(\zeta R)}{H_0(\zeta R)}, \tag{6}$$

$$p_{\text{inc}}^{(i)} = \sigma_0 \frac{\rho_f c_T^2}{\rho V_S^2} \left( \frac{1 - 2V_S^2/V_P^2}{1 - c_T^2/V_P^2} \right), \tag{7}$$

where $\sigma_0$ is the amplitude of the normally incident plane P-wave, $j = \sqrt{-1}$ the imaginary unit, $\omega$ the angular frequency, $k_r$ the radial wavenumber for a rigid, non-deformable fracture (a function of $L_0$), $\alpha_f$ the fluid velocity, $\alpha_{\text{eff}}$ the effective fluid velocity in the fracture (a function of $L_0$ and $Z$), and $R$ the borehole radius. $H_n$ denotes the Hankel function of the first kind of order

$n$, $\zeta$ the effective radial wavenumber (a function of $L_0$ and $Z$) and $\rho$ the density of the embedding background rock. $V_P$ and $V_S$ are the P-wave and S-wave velocity in the background rock, respectively. Note that $\sigma_0$ drops out of equation 4 due to the ratio of $p_t^{(i)}$ and $p_{\text{inc}}^{(i)}$.

When a tube wave propagating through a borehole encounters a fracture, fluid flow from the borehole into the fracture is triggered. This leads to reflection and transmission of tube waves. This process is described with the scattering potential $\phi_s$ in

the frequency domain:

$$\phi_s(z) = j\omega \sum_{i=1}^{N} \eta^{(i)} \delta(z - z_i), \tag{8}$$

where $\eta$ is the interface compliance given by:

$$\eta = -\frac{2\zeta}{R} \frac{L_0}{k_r^2 \alpha_f^2 \rho_f} \frac{H_1(\zeta R)}{H_0(\zeta R)}. \tag{9}$$

Note that the interface compliance differs from the fracture compliance. It linearly relates the velocity discontinuity $\Delta V$ across

the fracture to the acoustic pressure $p$: $\Delta V = j\omega \eta p$ (Minato and Ghose, 2017). Further details about the tube-wave generation and scattering potentials and the algorithm itself can be found in Minato and Ghose (2017).

For the forward operator $G$ as described so far, we assumed the fractures to be horizontally oriented. To account for arbitrary incidence angles, we have extended the above algorithm for the forward operator $G$ following the description given by Minato et al. (2017).

To improve the estimation of the fracture compliance $Z$, we have extended the forward operator of Minato and Ghose (2017) to include transmission losses of P-waves across fractures by using the angle-dependent transmission coefficient described by the linear slip theory (Schoenberg, 1980). Accordingly, the P- and S-wave reflection coefficients $R_P$ and $R_S$ as well as the P- and S-wave transmission coefficients $T_P$ and $T_S$ for an incoming P-wave are given by:

$$\begin{bmatrix} p_1 & \gamma_1 \cos(\psi_1) & p_2 & \gamma_2 \cos(\psi_2) \\ \gamma_1 \cos(\theta_1) & q_1 & \gamma_2 \cos(\theta_2) & -q_2 \\ -\sin(\theta_1) & -\cos(\psi_1) & \sin(\theta_2) - j\omega Z_T \gamma_2 \cos(\theta_2) & -\cos(\psi_2) + j\omega Z_T q_2 \\ \cos(\theta_1) & -\sin(\psi_1) & \cos(\theta_2) - j\omega Z_N p_2 & \sin(\psi_2) - j\omega Z_N \gamma_2 \cos(\psi_2) \end{bmatrix} \begin{bmatrix} R_P \\ R_S \\ T_P \\ T_S \end{bmatrix} = \begin{bmatrix} p_1 \\ \gamma_1 \cos(\theta_1) \\ \sin(\theta_1) \\ \cos(\theta_1) \end{bmatrix}, \tag{10}$$



where

$$\gamma_m \;\; = \;\; 2\rho_m V_{S_m} \sin(\psi_m), \tag{11}$$

$$p_m \;\; = \;\; \rho_m V_{P_m} - \gamma_m \sin(\theta_m), \tag{12}$$

$$q_m \;\; = \;\; \rho_m V_{S_m} \cos^2(\psi_m) - \frac{1}{2}\gamma_m \sin(\psi_m), \tag{13}$$

with the superscript $m$ being 1 for the medium above and 2 for the medium below the fracture. The angles $\theta_m$ and $\psi_m$ refer

to the P-wave and the S-wave reflection angles if the superscript $m$ is 1 and the corresponding transmission angles if the

superscript $m$ is 2. $Z_T$, $Z_N$, and $\rho$ denote the fracture compliance in the transverse direction (parallel to the fracture), the

fracture compliance in the normal direction (perpendicular to the fracture) and the density, respectively. Note that in this study,

we assume for simplicity that $Z = Z_T = Z_N$. We solve equation 10 for the four coefficients, but we only use the transmission

coefficient $T_P$ to reduce the amplitude of the P-wave after having crossed a fracture, because we do not consider reflections or

S-waves in this study.

In order to fit the observed data, we implemented the forward operator of Minato and Ghose (2017) such that the following

features are explicitly included: (1) Geometrical spreading of P-waves is accounted for by multiplying equation 7 with $1/z$.

Note that other attenuation mechanisms of the P-wave, besides geometrical spreading and transmission losses across fractures,

are neglected. (2) The algorithm assumes a uniform embedding background medium. To accommodate for P-wave velocity

changes above the considered borehole section, we introduce a variable source depth. This is an auxiliary parameter estimated

during the inversion. (3) The algorithm assumes an isotropic background medium. As the particle motion of a P-wave is

different compared to that of a tube wave in the elastic medium surrounding the borehole, the two wave types are sensitive

to the background medium properties in different directions. Therefore, taking anisotropy into account is important for fitting

observed data. We do this by estimating different effective isotropic shear moduli for the P-wave and for the tube wave. Thus,

the shear modulus $\mu$ in equation 5 becomes $\mu_t$, the tube-wave shear modulus.

Due to the non-linearity of the problem, we cannot infer the posterior PDF directly, but need to infer it by sampling the prior

PDF and the likelihood according to relation 1. For this, we chose to use a Markov chain Monte Carlo (MCMC) approach. This

algorithm walks randomly through the solution space accepting or rejecting proposed models $\mathbf{m}_{\mathrm{prop}}$, which are drawn from a

symmetric proposal distribution, with the Metropolis acceptance probability $\alpha$ (Metropolis et al., 1953):

$$\alpha = \min\left\{1, \frac{L(\mathbf{m}_{\mathrm{prop}}|\mathbf{d})p(\mathbf{m}_{\mathrm{prop}})}{L(\mathbf{m}_{\mathrm{cur}}|\mathbf{d})p(\mathbf{m}_{\mathrm{cur}})}\right\}, \tag{14}$$

where $\mathbf{m}_{\mathrm{cur}}$ is the model at the current location of the Markov chain. We use the DREAM$_{\mathrm{(ZS)}}$ algorithm (ter Braak and Vrugt,

2008; Laloy and Vrugt, 2012) to accomplish the sampling of relation 1 efficiently. DREAM$_{\mathrm{(ZS)}}$ allows for a fast convergence

towards the posterior PDF due to parallel and interacting Markov chains as well as a model-proposal scheme that uses a

database of previously accepted models to avoid sampling areas of low posterior probability and focusing on the interesting

areas of the solution space.





In the following, we apply the proposed inversion algorithm to hydrophone VSP data acquired at the Grimsel underground Test Site in the Central Swiss Alps. Previously, the viability and accuracy of the algorithm have been tested and verified on a variety of synthetic case studies, an example of which is shown in Appendix A.

## 3 Results

The VSP data, considered in the following, were recorded in crystalline rocks at the Grimsel Test Site in Switzerland using a 12-receiver hydrophone string with a receiver spacing of 1 m. In the course of the experiment, this hydrophone string was repositioned such that the recorded traces are separated by 50 cm. The borehole had a diameter of 14.7 cm. As a source, a single-handed 2 kg hammer was used at the wellhead, which excited frequencies between 0.1 and 4 kHz with a peak around 1.5 kHz. In this study, we consider a 20-m-long subsection, between 17 and 37 m depth, of this hydrophone VSP dataset

consisting of 41 receiver positions. This part of the borehole features three fractures at 23.5, 23.9 and 25 m depth, as evidenced from optical and acoustic televiewer data (Krietsch et al., 2018).

Preprocessing of the data included a gentle bandpass filter to suppress high-frequency noise, a static shift correction to remove positioning errors and a cosine taper to blank out the later arriving S-wave and associated tube waves. The thus processed data after this preprocessing are shown in Figure 1. The P-wave and the tube waves are clearly visible. However,

scattered tube waves, as described by equation 8, are too weak in amplitude and drop below the noise level. As the first and the second fracture are located closely together, the corresponding tube waves overlap, which poses a particular challenge for the inversion process. Before the data are supplied to the inversion algorithm, we separated the P-wave from the tube waves, applied a move-out correction to the P-wave and then calculated a mean trace. A time-gated version of this mean trace then serves as the estimate of the source wavelet.

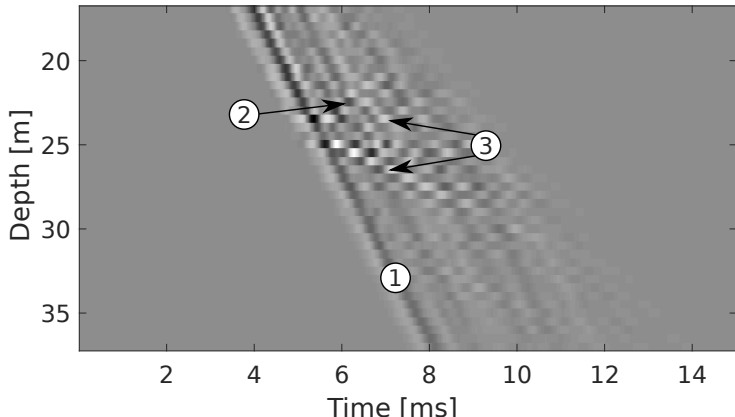

**Figure 1.** Observed hydrophone VSP data considered in this study. ① denotes the downgoing P-wave, ② the upgoing tube wave due to the fractures at 23.5 and 23.9 m depth, and ③ the up- and downgoing tube wave due to the fracture at 25 m depth. Note the amplitude decay associated with the P-wave.





**Table 1.** Unknowns of the inverse problem and their prior ranges subdivided by horizontal lines into three groups. The first group from the top comprises the background medium parameters, the second group the fracture parameters, and the third group algorithmic "tuning" parameters.

| | unknown | prior range | unit |
|---|---|---|---|
| background properties | bulk modulus of the background rock | $20 - 80$ | GPa |
| | shear modulus of the background rock for P-wave | $15 - 33$ | GPa |
| | shear modulus of the background rock for the tube wave | $2 - 50$ | GPa |
| fracture properties | aperture of first fracture | $10^{-4} - 10^{-2}$ | m |
| | aperture of second fracture | $10^{-4} - 10^{-2}$ | m |
| | aperture of third fracture | $10^{-4} - 10^{-1}$ | m |
| | compliance of first fracture | $10^{-15} - 10^{-10}$ | m/Pa |
| | compliance of second fracture | $10^{-15} - 10^{-10}$ | m/Pa |
| | compliance of third fracture | $10^{-15} - 10^{-10}$ | m/Pa |
| | depth of first fracture | $23.0 - 24.0$ | m |
| | depth of second fracture | $23.4 - 24.4$ | m |
| | depth of third fracture | $24.5 - 25.5$ | m |
| "tuning" parameters | source depth | $1.5 - 1.6$ | m |
| | tube-wave attenuation shift factor | $0.001 - 0.02$ | s |
| | tube-wave attenuation exponent | $0.0 - 1000.0$ | - |

For this problem with three fractures, we have 15 unknowns, which are specified in Table 1. Three unknowns are related to the background rock. These are the bulk and shear moduli of the formation and a separate shear modulus used for the tube waves. As outlined above, we use separate shear moduli for the P- and for the tube waves as a first-order approximation to account for anisotropy, which was estimated to be approximately 10% at the considered site (Wenning et al., 2018). The next nine unknowns are related to the fractures. For each of the three fractures, we estimate the hydraulic aperture, the compliance, and the depth. The forward solver also takes the fracture inclination into account. However, as tests on synthetic data showed that the fracture inclination cannot be inferred with high confidence, we assume that the inclination is known from televiewer data. The remaining three unknowns are algorithmic "tuning" parameters without any physical meaning. The first parameter of this group is the source depth. While the actual source location is known, we estimate the source depth for a fictitious homogeneous background medium to accommodate possible variations of the background medium parameters above the section under consideration. If the background rock is indeed homogeneous, the thus estimated source depth will correspond to the true source depth. The other two "tuning" parameters are used to emulate attenuation of the tube waves. As tube waves propagate along the borehole, they do not suffer from geometrical spreading as, for example, the P-wave does (Figure 1). However, tube waves are attenuated due to inelastic effects or scattering. To account for this, we dampen the tube waves using an exponential function defined by a shift factor, which specifies when the damping starts, and an exponent, which specifies the damping rate.





We have run our algorithm three times. Each time, three parallel Markov chains were used to explore the parameter space. The development of the root mean square error (RMSE) is plotted in Figure 2 for each Markov chain. Here, we weight the RMSE with the standard deviation of the data error. This means that, ideally, the weighted RMSE should converge to a value of one, with smaller values indicating that the data are over-fitted and larger values implying that not all the data can be explained by the proposed model. With the objective to force the algorithm to more extensively explore the posterior distribution, we

fix the standard deviation of the data error at a relatively high value, which is larger than corresponding estimates obtained in previous inversion runs. Figure 2 shows that all runs converge to a stable RMSE-value, which, as the data error is fixed at a high value, is smaller than one. Before reaching a stable RMSE, the algorithm explores the complete solution space in search of the posterior PDF. This is referred to as the burn-in phase. Subsequently, the algorithm is expected to have located the posterior PDF and to explore it in the course of the remaining iterations.

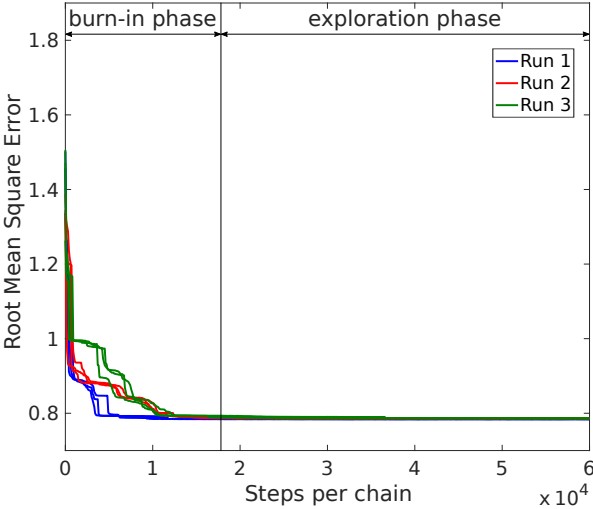

**Figure 2.** RMSE weighted by the standard deviation of the data error for the three inversion runs of the observed VSP data shown in Figure 1. As the estimate of the standard deviation of the data error is fixed at a high value, the RMSE drops below one. The vertical black line indicates the separation of the burn-in and the exploration phases, associated with the MCMC search of the parameter space.

In order to assess whether the Markov chains have converged sufficiently to allow for a reliable estimation of the posterior PDF, we calculate the so-called potential scale reduction factor $\hat{R}$ (Gelman and Rubin, 1992). Considering only the part of the Markov chains after burn-in, $\hat{R}$ compares the variance of the individual Markov chains with the overall variance of all the chains together. Usually, convergence is considered to be reached if $\hat{R}$ is smaller than 1.2 for all parameters. In this example, considering a burn-in phase of 30% of the complete chains, we get $\hat{R} < 2$ for most parameters, but only approximately a

third of the parameters reach $\hat{R} < 1.2$. Consequently, the posterior PDF has not been fully explored. Therefore, we do not plot posterior PDFs for the inferred parameters. Instead, we show the development of the Markov chains as a function of iteration number. Although proper convergence has not been achieved, the inferred models explain the data well. However,





other models, not sampled by the Markov chains, might explain the data well too. Hence, longer chains would be necessary to ensure a comprehensive exploration of the posterior PDF.

The acceptance rate specifies how many of the tested models are accepted. A too high acceptance rate generally implies that only models in the immediate neighborhood of the current model are explored while a too low acceptance rate means that computational resources are wasted by testing unrealistic models. Ideally, the acceptance rate ranges between 10 and 30%. In our case, it lies between 10 and 20% for runs one and two and around 5% for run three.

    The most interesting inferred parameters are the apertures and compliances of the fractures and, to a lesser extent, the
background rock properties. The development of these unknowns as a function of the number of iterations is plotted for all three runs in Figure 3. For the aperture of the first fracture (Figure 3a), the algorithm either finds a very large value of 10 mm (run 1) or a rather small one of less than 0.5 mm (runs 2 and 3). Interestingly, the opposite is the case for the second fracture (Figure 3b). Here, run 1 suggests a small fracture aperture and runs 2 and 3 a large one. As mentioned earlier, the first two fractures are very close together, at 23.5 and 23.9 m depth, respectively. Hence, the corresponding tube waves overlap. The
algorithm, thus, finds that one fracture must have a much larger aperture than the other, but it cannot determine which one is which. This leads to a bimodal posterior PDF featuring two equally probable modes. The estimated compliance values for these two fractures behave similarly (Figures 3d and 3e), although the difference between the runs is smaller.

    The vertical axis of the plots in Figure 3 represents the prior range. In the cases where the first or the second fracture is found to have a large aperture, the inferred value is actually located at the upper limit of the prior range. This means, that the
algorithm would propose even larger values if it were allowed to do so. We have not extended the prior range, because (1) even larger fracture apertures seem unrealistic and (2) the models found within this prior range are able to explain the data well.

    The posterior PDF for the estimates of the aperture of the third fracture is unimodal (Figure 3c). At the location of the third fracture, televiewer data (Krietsch et al., 2018) also indicate the presence of a larger shear zone. As we were not sure if the observed tube wave stems from the shear zone or the fracture, we extended the prior range by one order-of-magnitude to be
able to accommodate the complete shear zone. However, all three runs suggest a small aperture of less than 1 mm, which clearly indicates that the tube wave is generated by the fracture and not by the shear zone.

    For the bulk and shear modulus of the background (Figures 3g and 3h), we observe a similar behavior as for the fracture apertures of the first and the second fracture: If the bulk modulus is large, then the shear modulus is small and vice versa. Both parameters are constrained by two observables: (1) the P-wave velocity by the moveout of the P-wave and (2) the transmission
coefficient by the amplitude loss of the P-wave across fractures. However, these two observables are insufficient to constrain the background moduli adequately, thus leaving some degree of ambivalence in the final estimates. Conversely, the shear modulus used for the calculation of the tube-wave velocity is well constrained (Figure 3i), because there is no trade-off with other parameters.

    As the RMSE in Figure 2 is the same for all runs, the two modes of the posterior PDF identified by the algorithm explain
the data equally well. To further illustrate this, we compare in Figure 4 synthetic data for run 1 in blue (first fracture has a large aperture) and for run 2 in red (second fracture has a large aperture) with the observed data in black. Although we use a





**Figure 3.** Development of the most relevant unknowns for the three MCMC inversion runs of the observed VSP data shown in Figure 1: (a)-(c) apertures of the three fractures, (d)-(f) corresponding compliances, (g)-(i) elastic moduli.





**Figure 4.** Comparison between simulated (colored) and observed (black) data: (a) run 1 and (b) run 2





semi-analytic forward solver, which is inherently subject to a number of rather stringent assumptions, such as, for example, a homogeneous background medium, both synthetic datasets fit the observed data remarkably well.

## 4 Discussion

The acoustic and optical televiewer data of Krietsch et al. (2018) show that the first fracture has a larger aperture than the second one, confirming that the modal aperture distribution identified by run 1 is realistic. Conversely, the third fracture is too thin to infer its aperture from the televiewer data. This is also consistent with our findings, as we obtain a fracture aperture that is smaller than 1 mm. Concerning the fracture compliances, we can compare our results with those of Barbosa et al. (2019), who present corresponding estimates for the same borehole section based on full-waveform sonic log data. They

estimated fracture compliances which are approximately one order-of-magnitude higher than our results. Potential reasons for this difference might be that the full-waveform sonic data were measured at significantly higher frequencies ($\sim 20$ kHz) than our VSP data and that the fracture compliances tend to be frequency-dependent (e.g., Pyrak-Nolte, 1992; Nakagawa, 2013). Another difference between the two studies is the incidence angle. While Barbosa et al. (2019) assume normal incidence of the P-wave on the fractures, this study accounts for the dip angle of the fractures derived from televiewer data, which ranges from

$62°$ to $77°$ with regard to the horizontal.

A bit puzzling is the remarkably low estimate of the tube-wave shear-modulus of only about 6 GPa (Figure 3i). This parameter is very well constrained, as it is the only free parameter in equation 5, which may, however, be too simplistic for the following three reasons: (1) Equation 5 is derived in the low-frequency regime and its validity for higher frequencies is limited. (2) Attenuation of tube waves, as for example through scattering on the borehole tool or inside the damaged zone surrounding

the borehole, was not accounted for when estimating the tube-wave shear modulus. (3) Anisotropy is not taken into account completely. Thus, while the resulting tube-wave velocity is correct, as can be seen by the excellent fit between the observed and synthetic data, the corresponding shear modulus appears to be underestimated in order to correct for physical effects neglected in equation 5. Incorporating attenuation into the tube-wave velocity equation can be done by implementing equation 5-17 of White (1983) including the impedance of the borehole wall and accomodating anisotropy can be done by one of the methods

presented by Karpfinger et al. (2012). This, however, is beyond the scope of the present study.

From an inversion perspective, the most interesting point of these results is that two modes of the posterior PDF were identified, which showed that having the first fracture with a large aperture while the second fracture is thin, is similarly probable as the opposite scenario. Note that a deterministic approach would have provided only one result without any indication that there is another mode that can explain the data equally well, whereas our Bayesian approach clearly supplied us with both options.

This nicely demonstrates the value of stochastic inversion approaches.

A downside of our Bayesian approach is its enormous computational cost. Most of it is spent in the forward steps to simulate VSP data for a proposed model. We have optimized the forward simulation by parallelizing over frequencies. Still, one inversion run with three parallel Markov chains and 60'000 forward steps per chain took about 14 days to complete using one node (48 AMD Opteron 6174 processors at 2.2 GHz) of our cluster completely. However, the inversion would run three times faster





if each of the three Markov chains were run on a different node. We did not do this due to limited availability of resources. In any case, we argue that the computation time is well spent, since the results obtained are much more comprehensive than those that would be obtained through a deterministic inversion, as they allow, as explained above, to discover multiple modes of the posterior PDF. Furthermore, stochastic inversion approaches do not really depend on the starting model. This is in stark contrast to deterministic full-waveform inversion approaches, which require starting models whose forward response deviates from the forward response of the true model less than half a wavelength (Virieux and Operto, 2009).

We have decided not to estimate the source wavelet during the inversion process, although the corresponding algorithm was developed and successfully applied for synthetic test cases (Appendix A). The reason is, that the source wavelet of the observed data includes extensive reverberations and is, thus, extremely long and complicated. Estimating it as part of the inversion procedure would have required to more than double the amount of unknowns, which would have rendered the problem unnecessary complex.

An important limitation of our forward model, and indeed of virtually all fracture-based tube wave models, is that fracture aperture and compliance are correlated. This means that the inversion algorithm tends to compensate an overestimation of the fracture aperture by underestimating the fracture compliance. Therefore, we observe that a large fracture aperture for the first fracture is accompanied by a relatively small fracture compliance (Figures 3a and 3d). This is supposed to be mitigated in our approach, because the estimate of the fracture compliance is not only constrained by the tube-wave amplitude, but also by the reduction of P-wave amplitude when a fracture is crossed (Schoenberg, 1980). However, the transmission coefficients calculated for the estimated parameters are very close to 1 and, hence, the effect of this constraint is relatively weak. As the Markov chains are not oscillating all over the prior range and as the obtained values are reasonable, we can conclude that this compensation is rather limited.

Inspecting the difference between the observed and the forward modeled data shows that the largest discrepancies are found at the fracture locations. This indicates, that the transmission loss of the P-wave across fractures may not be reproduced properly in the synthetic data. However, as this affects only the P-wave around the fracture locations, the impact on the RMSE are limited. A possible way to improve this issue might be to define a weighting function that peaks at the fracture locations to force the algorithm to obtain a better data fit at these locations and, thus, find a more accurate transmission coefficient. The downside of this is, however, that the weights are new "tuning" parameters that need to be adjusted through a time-consuming process, which was not feasible to accomplish in the scope of this study.

Limitations of our implementation of the forward operator are its inability to account for scatterers or geological facies changes. If corresponding effects are present in the data, they need to be filtered out prior to inversion. Similarly, changes in the P-wave velocity are not taken into account. If these are present, the data needs to be cut into smaller pieces with constant P-wave velocity. Changes in P-wave velocity above the considered borehole section are taken into account by virtually shifting the source depth. The algorithm is also not able to take S-waves and corresponding tube waves into account. In our dataset, events of this kind were indeed present and needed to be muted before applying our inversion algorithm to it.





# 5   Conclusions

We have developed a Bayesian MCMC full-waveform inversion algorithm based on a semi-analytical forward solver to si-
multaneously infer the aperture and compliance of individual fractures from corresponding tube-wave data. We mitigate the
correlation between fracture aperture and compliance by constraining the fracture compliance by two independent observables:
(1) the tube-wave amplitude relative to the P-wave amplitude and (2) the amplitude loss of the P-wave across a fracture. The
algorithm was applied to a field dataset acquired in crystalline rock at the Grimsel Test Site in Switzerland. The subsection
of the VSP dataset considered contained three fractures, of which two are very close together. The algorithm identified two
equally probable modes in the posterior PDF: Either the first fracture features a large aperture and the second fracture as small
one or vice versa. In other words, from the information provided, the algorithm can determine that one fracture is larger than
the other, but it cannot determine which one is thick and which one is thin. The identification of these two modes clearly illus-
trates a major advantage of stochastic inversion algorithms as compared to their deterministic counterparts. The latter would
not have identified these two modes and would have provided just one of the two possible solutions. The inferred apertures are
consistent with televiewer data and the inferred compliances are roughly in the same range as those derived from sonic logs
at the same site. The data fit is remarkably good, especially when considering the semi-analytical nature of the forward solver
and the inherent assumptions it relies on as well as the rather complex character of the observed hydrophone VSP data.

*Code availability.*    The forward solver can be download from https://github.com/rockphysicsUNIL/tube_wave_forward_solver.

# Appendix A:  Synthetic example with real noise

Before applying our inversion algorithm to observed data, we have run tests on synthetic data to ensure that the algorithm
performs as expected. As in these experiments the same forward solver was used for the generation and the inversion of the
data, the corresponding results only allow to draw conclusions with regard to the inversion algorithm itself, but not with regard
to the information content of the data. The test case shown here features two fractures at 10 and 19 m depth. The receiver
spacing is 1 m. To make this synthetic study more pertinent and challenging, we contaminated the dataset with actual ambient
noise from a corresponding field dataset at the Grimsel Test Site in Switzerland. The resulting data are plotted in Figure A1a.

This synthetic test differs from the field-data example shown above in two ways: (1) It uses as a forward solver the algorithm
proposed by Minato and Ghose (2017) and Minato et al. (2017) without taking transmission losses, geometrical spreading
for P-waves, velocity changes above the considered borehole section or anisotropy into account, because these were also not
included when the data were created. (2) While the wavelet is inferred based on a mean trace for the field data, we treat it as
unknown and, thus, estimate it in the synthetic example. We do this, by inferring the coordinates of six pilot points from which
we obtain the wavelet by a shape-preserving piecewise cubic interpolation (Hunziker et al., 2019).

The inversion was run once with three parallel Markov chains. Figure A2 shows the estimate of the hydraulic fracture
aperture and the mechanical compliance for the two fractures as a function of the number of forward simulation steps. For all





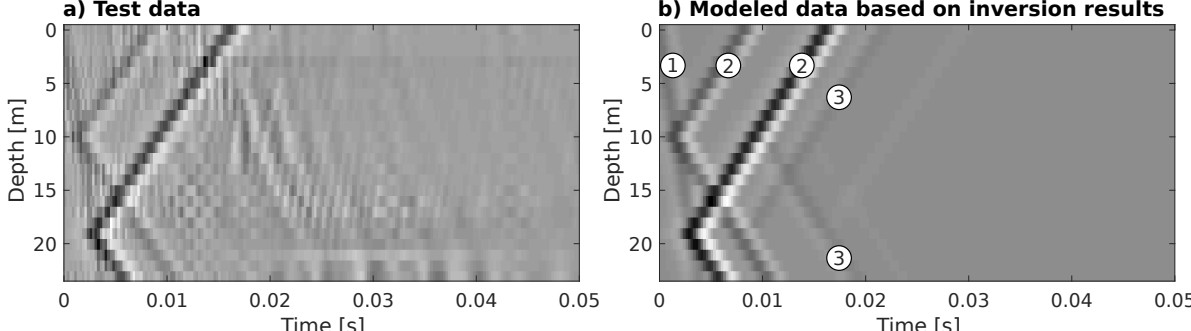

**Figure A1.** a) Synthetic test data featuring two fractures at 10 and 19 m depth contaminated with ambient noise from observed hydrophone VSP data acquired at the Grimsel Test Site in Switzerland; b) simulated data based on an inferred model at the end of a Markov chain. ① denotes the direct P-wave, ② the tube waves generated at the fractures, and ③ the tube waves reflected at the fractures.

four unknowns, the three chains converge nicely to the true value. This shows, that the algorithm works properly even when

the data are contaminated with correlated, realistic noise.

Simulated data based on a model proposed at the end of the first Markov chain are reproduced in Figure A1b and agree very well with the input data (Figure A1a). Note that besides the direct P-wave (①) and the tube waves generated at fractures (②), also the tube waves reflected at fractures (③) are visible. The latter are visible neither in the noise-contaminated input data nor in the actual field data.

*Acknowledgements.* This work has been completed within the Swiss Competence Center on Energy Research – Supply of Electricity, with the support of Innosuisse.

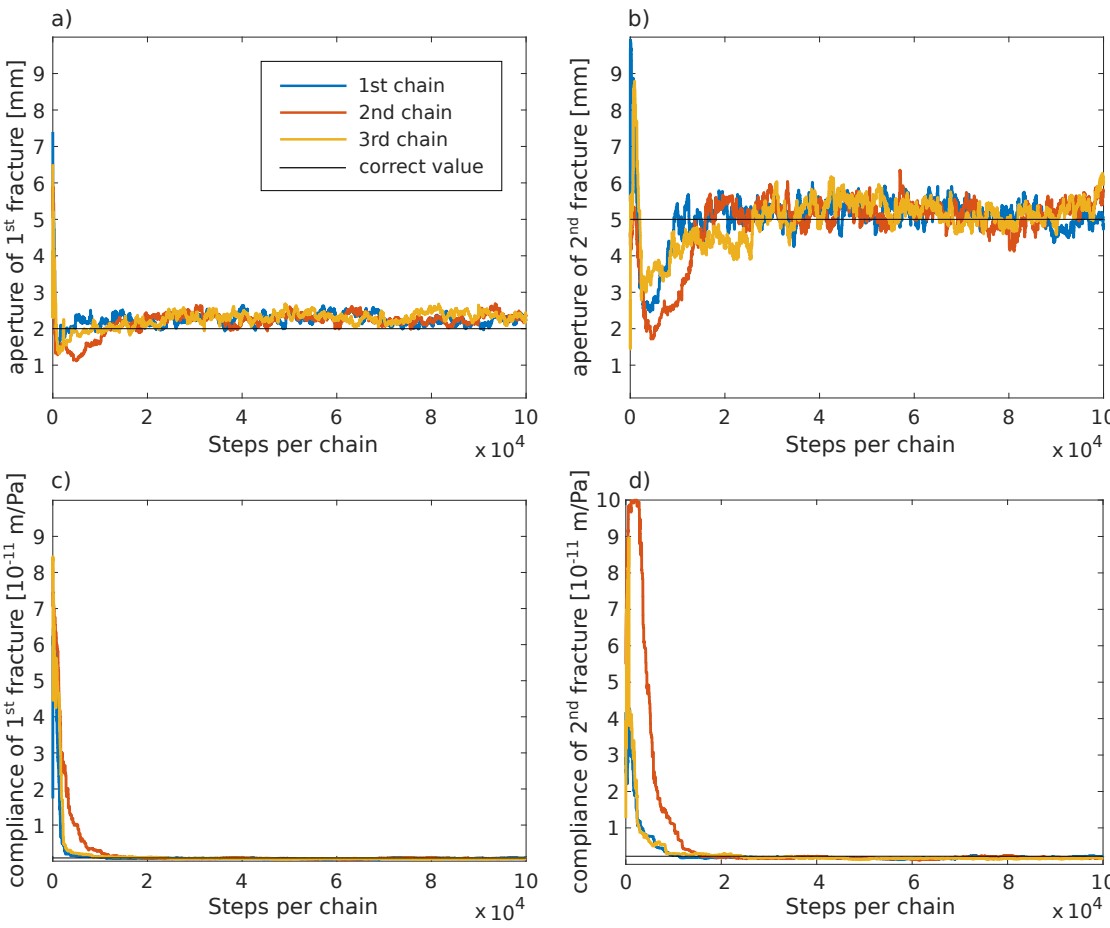

**Figure A2.** Estimates of (a-b) the aperture and (c-d) the compliance of the two fractures as functions of the number of forward modeling steps. the horizontal black lines denote the corresponding values used to generate the synthetic data shown in Figure A1a.

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
