# Peer review of "Bayesian full-waveform inversion of tube waves to estimate fracture aperture and compliance"

_Solid Earth, 2019_

## Referee Comment (RC1) · Anonymous Referee #1 · 5 Jan 2020

The paper is interesting and generally well written. The authors present a method to successfully estimate fractures apertures and compliances using Bayesian Full Waveform Inversion of VSP tube-wave data. However, since the method can not clearly discriminate between the aperture of fractures that are relatively close to each other, do the authors think that the method could be successfully applied in complex geological settings such as in carbonates that are usually characterized by the presence of several fractures?

Minor corrections:

Figure 1; please consider improving the resolution so that the different phases can be followed easily.

[Figure]

Page 6 Line 135: remove "thus".

Page 6 Line 158: "A time-gated version of...." please specify the window size.

Page 8 Line 175: change " We have run" to we ran.

Page 9 Line 193: Please consider changing "might explain the data well too" to "might equally explain the data".

Page 14 Line 300: " ... and the second fracture as small one..." replace as with a.

---

## Referee Comment (RC2) · Anonymous Referee #2 · 9 Jan 2020

This paper proposed a new inversion algorithm of tube waves and detailed analysis. However, the logic of this paper is not clear and strange.

(1) There is indeed numerical test. Yet the numerical test is quite just analysis, and there is no real data tests.

(2) Take figure 4 as example. How is the simulated result generated in detail? Throughout the paper, what does "synthetic" mean? Concrete inversion results are not found.

(3) why there is a numerical example in the appendix?

Separately, add the reference where formula (4) came from.

---

## Referee Comment (RC3) · Anonymous Referee #3 · 13 Feb 2020

The authors present a novel approach for Bayesian full-waveform Inversion of VSP tube waves with the aim of estimating fracture aperture and compliance. The manuscript is well written and interesting to read. The choice of the research methodology is appropriate and it supports the research objective. The authors validate the proposed approach using both synthetic and experimental data.

I have a couple of comments and questions:

- Could you please elaborate more on the choice of inversion parameters and their influence on the results, e.g. number of Markov chains, number of runs.

- I understand that the number of fractures is a known parameter. How do you define the total number of fractures for inversion and what is the smallest fracture aperture

that you can consider?

- It is known that the tube wave reflection and transmission is not only generated through a fracture intersecting a borehole, but it can be also caused by borehole diameter changes (i.e. washouts). Can your inversion algorithm account for this effect?

- I'd suggest to include the synthetic example in the main body rather than in the Appendix. This will improve the readability and understanding of the method.

- The authors state that the inversion results are consistent with the televiewer data and refer to Krietsch et al. (2018) many times. Which figure in Krietsch et al. (2018) is showing the interval selected for the Bayesian inversion? Could you please include it in your manuscript for clarity?

- "The inferred apertures are consistent with televiewer data and the inferred compliances are roughly in the same range as those derived from sonic logs at the same site." Please be more precise, what are the apertures and compliances values derived from the televiewer data and sonic logs. What is the vertical resolution of the televiewer data?

---

## Author Comment (AC1) · 12 Mar 2020

Response to reviewer 1

Thank you for your comments. Please find our detailed responses to your comments below in red font.

The paper is interesting and generally well written. The authors present a method to successfully estimate fractures apertures and compliances using Bayesian Full Waveform Inversion of VSP tube-wave data. However, since the method can not clearly discriminate between the aperture of fractures that are relatively close to each other, do the authors think that the method could be successfully applied in complex geological settings such as in carbonates that are usually characterized by the presence of several fractures?

Based on extensive tests, we are convinced that the method presented in this paper will also be useful in even more complex geological environments than discussed in this study. However, the method is not sensitive to the complex pore space often encountered in carbonates, as these micro cracks and pores do not produce tube waves. Although, estimates of fracture apertures of individual, closely spaced fractures are not possible, the method can still provide an effective fracture aperture distribution of a package of fractures. We have added some text to the conclusions to clarify this. Furthermore, and this is in our view one of the key findings of this paper, our algorithm will not produce the illusion of being able to discern different fracture apertures as would be the case when using a deterministic inversion approach. Instead, our algorithm infers all statistical modes that are probable given the data.

Minor corrections:

- Figure 1; please consider improving the resolution so that the different phases can be followed easily.
  Given that the temporal resolution of the data is quite high, we assume that you refer to depth resolution. We do indeed agree that the figure does not look great. However, these are real data that were measured with a depth resolution of one trace per 0.5 m, which is a typical value for surficial high-resolution VSP surveys. This inherently limited depth resolution is reflected in the figure and cannot be improved without heavily interpolating, and thus, biasing the original data. However, it also nicely illustrates that the proposed inversion algorithm can handle such data well, despite the seemingly low resolution with regard to depth.

- Page 6 Line 135: remove "thus".
  The only "thus" on page 6 is on line 153. We assume you refer to this one and removed it.

- Page 6 Line 158: "A time-gated version of...." please specify the window size.
  The window-length is 10 ms. We added this information to the manuscript.

- Page 8 Line 175: change " We have run" to we ran.
  Done.

- Page 9 Line 193: Please consider changing "might explain the data well too" to "might equally explain the data".
  Done.

- Page 14 Line 300: " ... and the second fracture as small one..." replace as with a.
  Done.

---

## Author Comment (AC2) · 12 Mar 2020

Response to reviewer 2

Thank you for your comments. Please find our detailed responses to your comments below in red font.

This paper proposed a new inversion algorithm of tube waves and detailed analysis. However, the logic of this paper is not clear and strange.

We assume that with "logic of this paper" you refer to the ordering of the sections. To address this issue, we have moved the section presenting the synthetic example into the main body of the paper. Now, the paper presents first the method and all the theory related to it, then demonstrates the viability of the proposed algorithm based on a synthetic example, followed by a real-data example, which features a detailed presentation of all the practical aspects of the algorithm. Finally, we address advantages and disadvantages as well as limitations of the algorithm in the discussion section. We think this revised structure now guides the reader in a smooth and logic way through the paper.

(1) There is indeed numerical test. Yet the numerical test is quite just analysis, and there is no real data tests.

There are two examples in the paper: a synthetic example and a real-data example. We use these examples to analyze the viability and the performance of our proposed algorithm. It is not clear to us, what you suggest us to do by saying "the numerical test is quite just analysis". Furthermore, we would like to draw your attention to the results section, which mainly consists of a real-data test.

(2) Take figure 4 as example. How is the simulated result generated in detail? Throughout the paper, what does "synthetic" mean? Concrete inversion results are not found.

For the simulated data, we took the model obtained by our inversion (the last model of the third Markov chain of run 1 and run 2) and generated synthetic data using our forward solver described in the methods section. To further clarify this, we have added some explanatory text in the results section when describing this figure.

The term synthetic refers to any kind of data that is simulated, and thus, not real, measured or observed. So, the forward solver creates synthetic data that are in the course of the inversion compared with the observed data. What we treat as "observed" data in the synthetic example are in fact synthetic data, as they are also simulated.

Concrete inversion results are shown in Figure 5 (the figure number refers to the new manuscript; this was Figure 3 in the old manuscript). These plots show the inferred model parameters for all Markov chains, but we do not show posterior PDFs due to limited convergence as explained between lines 213 and 222 (refering to the updated manuscript). The results are presented in the results section and discussed in the discussion section.

(3) why there is a numerical example in the appendix? Separately, add the reference where formula (4) came from.

We have moved the numerical example to the results section.

Equation 4 is equation 22 in Minato and Ghose (2017). An additional reference has been added to the manuscript to clarify this.

---

## Author Comment (AC3) · 12 Mar 2020

Response to reviewer 3

Thank you for your comments. Please find our detailed responses to your comments below in red font.

The authors present a novel approach for Bayesian full-waveform Inversion of VSP tube waves with the aim of estimating fracture aperture and compliance. The manuscript is well written and interesting to read. The choice of the research methodology is appropriate and it supports the research objective. The authors validate the proposed approach using both synthetic and experimental data.

I have a couple of comments and questions:

- Could you please elaborate more on the choice of inversion parameters and their influence on the results, e.g. number of Markov chains, number of runs.
  The number of Markov chains was chosen as a trade-off between exploring the solution space as comprehensively as possible and keeping the computational costs manageable. More Markov chains allow for a more exhaustive exploration of the solution space in search of the posterior probability density function, but require more computational resources. If vast computational resources are available, many Markov chains can be run in parallel. As this was not the case for us, we chose the minimal number of Markov chains that still allows for an adequate exploration of the solution space.
  The number of runs was chosen based on a similar reasoning. For infinite chains, our Bayesian inversion should always find the posterior probability density function. However, as the chains are finite, there is a certain risk that the algorithm ends up in a local minimum. If the latter is the case, several different runs would lead to different results. So, to ensure that the posterior probability density function has been found, multiple runs are necessary, at least two, ideally more. Three runs was the maximum that could be done within a reasonable amount of time with the computational resources at hand.
  The section presenting the real-data experiment has been updated to explain our choice of these parameters.

- I understand that the number of fractures is a known parameter. How do you define the total number of fractures for inversion and what is the smallest fracture aperture that you can consider?
  To determine how many fractures are included in the inversion, the measured data are inspected. For each clearly discernible tube-wave event, a fracture is included in the inversion. The origin of the tube wave thereby defines the depth of the corresponding fracture. If available, as in our study, an optical televiewer log can be consulted to avoid associating non-fracture-related tube-wave events with fractures or identifying cases where multiple, closely spaced fractures create one big tube-wave event. The manuscript has been adapted to state that the fractures have been identified through visual inspection of the seismic data while also considering the available televiewer data.
  As our forward solver is semi-analytical, there is no limit to fracture apertures that we can consider. However, there is a spatial limitation with respect to the depth sampling. In our case, we sample along the borehole with a spacing of 0.1 m. As fractures have to be located at these sampling points, two adjacent fractures cannot be closer together than 0.1 m. In order to model smaller distances between fractures, the depth sampling needs to be densified, thus, slowing down the algorithm significantly, unless the considered borehole section is shortened. A statement explaining the link between depth sampling and minimal distance between fractures has been added to the methods section.

- It is known that the tube wave reflection and transmission is not only generated through a fracture intersecting a borehole, but it can be also caused by borehole diameter changes (i.e. washouts). Can your inversion algorithm account for this effect?
  Our forward solver does not take these secondary sources of tube waves into account (except if fractures act as secondary sources). We assume that borehole diameter changes are spatially separated from fractures. In that case, such an event will only increase the data misfit, but will not affect the estimate of fracture-related parameters. In contrast, if a fracture is close to a borehole diameter change, the tube wave caused by the fracture will be affected by the borehole diameter change and, consequently, also the fracture-parameter estimates will be affected. We have added some text in the methods section clarifying that other sources for tube waves, besides fractures, are neglected in our forward solver.

- I'd suggest to include the synthetic example in the main body rather than in the Appendix. This will improve the readability and understanding of the method.
  We have followed your suggestion and adapted the manuscript accordingly.

- The authors state that the inversion results are consistent with the televiewer data and refer to Krietsch et al. (2018) many times. Which figure in Krietsch et al. (2018) is showing the interval selected for the Bayesian inversion? Could you please include it in your manuscript for clarity?
  The paper by Krietsch et al. (2018) is a data description for a dataset that is freely accessible here:
  `https://www.research-collection.ethz.ch/handle/20.500.11850/243199`
  The compressed folder "3D static geological Model.zip" contains the following file:
  `/3D static geological Model/05_GeophysicalBoreholeLogs/INJ2/INJ2_structures.txt`
  which lists the estimated fracture apertures from televiewer data for each fracture. We compared our results with these data. As there is no figure in the paper by Krietsch et al. (2018) that shows these data, we cannot reproduce it in our paper.

- "The inferred apertures are consistent with televiewer data and the inferred compliances are roughly in the same range as those derived from sonic logs at the same site." Please be more precise, what are the apertures and compliances values derived from the televiewer data and sonic logs. What is the vertical resolution of the televiewer data?
  The first paragraph of the discussion section has been extended to state the values explicitly and how we interpret them. The vertical resolution of the televiewer data is 0.21 mm. This information has also been added to the discussion section of the manuscript.